# Co-infection of HIV or HCV among HBsAg positive delivering mothers and its associated factors in governmental hospitals in Addis Ababa, Ethiopia: A cross-sectional study

**Mebrihit Arefaine Tesfu** *, **Nega Berhe Belay, Tilahun Teklehaymanot Habtemariam**

Aklilu Lemma Institute of Pathobiology, Addis Ababa Universty, Addis Ababa, Ethiopia

* mebrihitarefaine@gmail.com

## Abstract

### Background

Blood borne viral infections such as Hepatitis B virus (HBV), Hepatitis C virus (HCV), and Human Immunodeficiency virus (HIV) cause substantial mortality and morbidity worldwide. Viral hepatitis during pregnancy is closely related to high risks of maternal and neonatal complications. In Ethiopia, only a little information is available on co-infection of HCV or HIV among Hepatitis B surface Antigen (HBsAg) positive pregnant mothers. Thus, the study aimed to determine HIV or HCV co-infection and associated risk factors among HBsAg positive delivering mothers.

### Method

A health facility-based cross-sectional study was conducted in five governmental hospitals in Addis Ababa among 265 HBsAg positive delivering mothers in the year 2019 and 2020. A purposive sampling technique was used to select the study participants. Structured questionnaires and laboratory test results were used to collect the data. SPSS version 20 software was used to enter and analyze the data. Multivariable logistic regression was used to identify independent predictors of HIV or HCV co-infections.

### Results

Of the HBsAg positive delivering mothers, 9 (3.4%) and 3 (1.1%) were co-infected with HIV and HCV, respectively. None of them were with triplex infection. All of the socio-demographic characteristics were not significantly associated with both HIV and HCV co-infections. Mothers who had a history of sexually transmitted diseases (STDs) were 9.3 times more likely to have HBV-HIV co-infection (AOR = 9.3; 95% CI: 1.84–47.1). Mothers who had multiple sexual partners were 5.96 times more likely to have HIV co-infection (AOR = 5.96; 95% CI: 1.074–33.104). The odds of having HBV-HIV co-infection were 5.5 times higher among mothers who had a history of sharing shavers, razors, and earrings (AOR =

**Data Availability Statement:** All relevant data are within the paper.

**Funding:** The author(s) received no specific funding for this work.

**Competing interests:** The authors have declared that no competing interests exist.

5.5;95% CI: 1.014–29.69). HCV co-infection was not significantly associated with any of the potential risk factors.

## Conclusion

This dual infection rate of HBsAg positive delivering mother with HIV or HCV indicates that a substantial number of infants born in Ethiopia are at high risk of mother-to-child transmission (MTCT) of HBV, HIV, and HCV. Thus, all pregnant mothers need to be screened for HBV, HCV, and HIV during antenatal care, and also need implementation of prevention mechanisms of MTCT of these viral infections.

## Background

Blood-borne viral infections such as HBV, HCV, and HIV cause substantial mortality and morbidity worldwide [1]. HBV infection occurs globally and constitutes a major public health problem. It infects over 20 million people globally every year, and there are around 350–400 million chronic carriers. More than 1.2 million deaths occur annually from HBV-related disease, making it the 10th leading cause of death, and the second most common cause of cancer deaths after tobacco, globally [2, 3]. HBV is an infectious disease, and it mainly transmits through mother-to-child, skin and mucous membrane infections by contaminated blood or body fluids, sexual contacts, and injection drug abuse. In addition, body tattooing, ear piercing, acupuncture, dialysis, and even using a syringe can be the source of infection [4]. HBV acquisition in adulthood commonly leads to acute resolved infection and immunity. But, perinatal/neonatal HBV infection more likely leads to chronic infection and its long-term disease risks [5].

Worldwide the prevalence of HCV infection in pregnant women and children has been estimated at 1–8% and 0.05–5%, respectively. Although, direct percutaneous inoculation is the most efficient mode of transmission of HCV, sexual, household, occupational, and vertical transmission may also be important [6–8]. MTCT of HCV increases to 4–25% times if the mother is also HIV positive [9].

Over the last decades, HIV infection has been one of the largest public health challenges, especially in low and middle-income countries [7]. Among the HIV positive patients, 2–4 million are estimated to have chronic HBV co-infection, and 4–5 million are co-infected with HCV [10]. HIV in pregnancy has adverse outcomes to maternal and fetal health and also to health workers at times of delivery [8].

HBV, HIV, and HCV have similarities in mode of transmission, but there are slight variations in the mode of transmission of these viruses [10–12]. Unlike HIV, HBV is not transmitted by breastfeeding, furthermore, child to child transmission is common for HBV but not for HIV. HBV is 50–100 times and 10 times more infectious than HIV and HCV, respectively [7]. The presence of higher concentrations of HBV in body fluids of persons with acute or chronic HBV infection than persons with HIV infection may cause HBV more infectious than HIV [10].

Co-infection of both HBV and HCV with HIV is associated with low CD4 count, accelerated liver disease progression, higher mortality, and MTCT of the viruses [7, 13–15]. Additionally, the progression rate and complications such as liver fibrosis, cirrhosis, end-stage liver disease, hepatocellular carcinoma (HCC), and mortality due to liver pathology arising from HBV infection are accelerated in patients with HIV co-infected patients than in patients with

HBV infection alone [7, 16]. The clinical management of individuals co-infected with those viruses is challenging [17].

Available data suggest that, in resource-rich settings, approximately 10% of the HIV infected population have chronic HBV infection and around a third have chronic HCV infection [13, 16]. However, wide regional variations are observed with co-infection of HIV and HBV prevalence rates estimated to be 5–10% in areas such as North America, Europe and Australia compared to higher prevalence rates of 20–30% in areas such as Sub-Saharan Africa and Asia [16].

HBV/HIV co-infection rates of 12.9% [18] and 16% [19] were reported in people living with HIV in Cameroon. Besides, HBV/HIV co-infection rates of 11.8% [20] and 3.1% [21] were reported in pregnant mothers in Nigeria and South Africa, respectively. HCV/HIV co-infection rates of 0–33% were reported in pregnant mothers in Nigeria [22–24].

HBV/HIV co-infection rates of 19–40% were reported among pregnant mothers according to different studies conducted in Ethiopia [7, 25, 26]. One hundred percent of HCV positive pregnant mothers were also co-infected with HIV in Atat Hopsital, Southern, Ethiopia [7], and 2.9% of HCV/HIV co-infection was reported in pregnant mothers in East Wollega Zone, West Oromia, Ethiopia [9]. None of the pregnant mothers were co-infected with HBV and HCV [27, 28]. Several studies were conducted to determine co-infection of HBV/HIV, HBV/HCV, or HIV/HCV among the general population, general pregnant mothers, or HIV infected pregnant mothers [7, 13, 16, 18–28]. But, only a few studies have been conducted to determine co-infection of HIV or HCV and related risk factors among HBsAg positive delivering mothers globally and particularly in Ethiopia. The study aimed to assess the co-infection of these viruses among HBsAg positive delivering mothers and factors associated.

## Methods and materials

### Study design and period

A health facility based cross -sectional study design was employed from January 2019 to December 2020.

### Study site and population

The study was conducted in Addis Ababa, the capital city of Ethiopia, with an estimated population of 4,591,983 million in 2019 [29]. The city has a higher population of female occupants compared to male occupants. In the city, there are 14 governmental hospitals and 103 health centers. In addition to this, there are many private hospitals and clinics which provide health service for the community of the city and patients from other parts of the country. The study was conducted in 5 of the governmental hospitals that have maternal and child health care services. The hospitals were Zewditu Memorial Hospital, Gandhi Memorial Hospital, Yekatit 12 Hospital Medical College, Armed force Hospital and Menelik II referral Hospital. The study population was HBsAg positive delivering mothers who attended in the selected governmental hospitals in Addis Ababa.

### Sample size determination and sampling technique

A single population proportion formula was used. The sample size (n) of this study was determined using 22.2% of HBV/HIV co-infection in pregnant women in Addis Ababa [26] and giving any particular outcome to be with 5% marginal error and 95% confidence interval. Based on this assumption, the actual sample size for this study was computed using one sample

population proportion formula indicated below

$$n = (z\partial/2)^2 p(1-p)/d^2$$

Where n = sample size, P = 0.222, d = 0.05(5% error of margin), z $\partial/2$ = 1.96 (standard normal probability for 95% CI) n = 265

After identifying governmental health facilities that have maternal and child health services, study health facilities were selected using the simple random sampling method. A purposive sampling technique was employed to select the study participants.

## Eligibility

**Inclusion criteria.** All HBsAg positive delivering mothers who attended the selected hospitals in Addis Ababa and those who were volunteer to participate and give informed consent were included.

**Exclusion criteria.** Delivering mothers who had communication problems were excluded.

## Study variables

**Dependent variables.** Co-infection of HIV among HBsAg positive delivering mothers and co-infection of HCV among HBsAg positive delivering mothers were the dependent variables.

**Independent variables.** Age, marital status, educational status, occupation, religion, and gravidity were socio-demographic independent variables. History of blood transfusion, STDs, abortion, surgical procedure, dental procedure, tattooing, ear piercings, nose piercing, home delivery by traditional birth attendants, multiple sexual partners, female circumcision, hospital admission, Presence of known hepatitis B infected person in a family, sharing shavers, razors, or earrings (at homes, beauty salon or barbershops), sharing toothbrushes, history of jaundice and contact with jaundice patients were the potential risk factor independent variables.

## Data collection

To collect the data, questionnaires were developed by the study group in English and then translated to Amharic and back to English and interview was done by the data collectors. Laboratory test results of the blood samples were also used.

## Data collection procedures and quality assurance

Delivering mothers who were diagnosed HBsAg positive during screening for HBV as routine antenatal care service were selected based on the status on their medical records. To get information on socio-demographic characteristics of the respondents and risk factors associated with co-infection of the viruses, pre-designed and pretested structured questionnaires were used. Ten midwives and medical laboratory technologists working in the selected hospitals were collected the data. Five supervisors with a second degree in health-related fields who recruited based on their experience in data collection and supervision participated. Two days of training were given on the objective of the study, obtaining consent, confidentiality of the information, and data collection procedures for the data collectors and supervisors. Questionnaires were carefully designed and pre-tested with individuals' equivalent to 5% of the calculated sample size in Ras Desta Damtew Hospital. Questionnaires were slightly adjusted after pretested results show a lack of clarity. To determine the co-infection of the viruses, 5 ml of venous blood was collected based on the standard collection procedure and placed in

ethylene-diamine-tetra-acetic acid (EDTA) tubes. These tubes were labelled with unique identification number and processed at the time of collection. The supervisors and the principal investigator supervised the data collection process.

## Laboratory procedures

The blood samples taken from the individuals were centrifuged at 3000 revolutions per minute (RPM) for at least 10 minutes at room temperature. Then the plasma was tested for HBsAg (to re-check the status) and anti-HCV to determine HCV infection using separated rapid test Cassettes (Nantong diagnosis biotechnology co.Ltd P.R.china), which have specificity and sensitivity of greater than 99%.Testing was done following the manufacturer's protocol. The HIV status of the delivering mothers was used from the antenatal care records since the test is performed on a routine basis.

## Data management and analysis

The generated data was cleaned, coded, and uploaded into a computer using SPSS version 20.0 statistical software for analysis and interpretation. Descriptive values were expressed as the frequency, percentage, and mean ± standard deviation (SD). Logistic regression analysis was implemented to explore and determine the relationship of predictors on outcome variables. Variables significant at $p < 0.25$ with the dependent variable were selected for multivariable analysis. Odds ratio with 95% confidence level was computed, and a significant association was declared at $p < 0.05$.

## Ethical consideration

Ethical clearance and approval were obtained from the Institutional Review Board of Aklilu Lemma Institute of Pathobiology, Addis Ababa University, and the Addis Ababa Health Bureau. Permission to carry out the study was obtained from the health facilities. After explaining the purpose, written informed consent was obtained from the delivering mothers. Moreover, confidentiality was assured for all the information provided, and the personal identifiers were not included in questionnaires. Results were reported to physicians for treatment management and prevention of MTCT of these viral infections.

## Results

### Socio-demographic characteristics of the study participants

A total of 265 HBsAg positive delivering mothers who attended governmental hospitals in Addis Ababa were included. The mean age of the study participants was 28.04 years with a SD of 4.62. The majority of the delivering mothers (43.1%) were in the age group of 25–29 years. Most (92.5%) of the delivering mothers were married. About 45.3% of the mothers were housewives, followed by 19.6% who were private employees and 17.7% who were government employees. Respondents were predominantly Orthodox Christians (71.7%) followed by Muslims (16.6%) and Protestants (9.8%). More than half (60%) were multigravida mothers (Table 1).

### Prevalence of HIV co-infection among HBsAg positive delivering mothers

Out of the 265 HBsAg positive delivering mothers, 9 (3.4%) were co-infected with HIV. Of those, 77.8% were married and multigravida mothers, and 66.7% were between25-29 years old and orthodox Christians. There were no significant differences in HIV co-infection prevalence among all socio-demographic characteristics of the study participants (Table 2).

**Table 1. Socio-demographic characteristics of HBsAg positive delivering mothers in governmental hospitals in Addis Ababa, 2019–2020.**

| Variable | Category | Number | Percentage |
|---|---|---|---|
| Age | <20 | 4 | 1.5 |
| | 20–24 | 61 | 23 |
| | 25–29 | 115 | 43.4 |
| | 30–34 | 60 | 22.6 |
| | >35 | 25 | 9.4 |
| Marital status | Married | 245 | 92.5 |
| | Single | 12 | 4.5 |
| | Divorced | 8 | 3 |
| Educational status | No formal education | 28 | 10.6 |
| | Primary level (1–8) | 98 | 37 |
| | Secondary level (9–12) | 75 | 28.3 |
| | College diploma and above | 64 | 24.2 |
| Occupation | Government employee | 47 | 17.7 |
| | Private employee | 52 | 19.6 |
| | Self- employee | 27 | 10.2 |
| | House wife | 120 | 45.3 |
| | Others | 19 | 7.2 |
| Religion | Orthodox Christian | 190 | 71.7 |
| | Muslim | 44 | 16.6 |
| | Protestant | 26 | 9.8 |
| | Others | 5 | 1.9 |
| Gravidity | Primigravida | 106 | 40 |
| | Multigravida | 159 | 60 |

## Risk factors associated with HIV co-infection among HBsAg positive delivering mothers

Most (88.9%) of the HIV co-infected mothers had a history of ear piercings, and 44.4% of them had a history of STDs, previous abortion, and female circumcision. About 7 (77.8%) and 6 (66.7%) of them had a history of having multiple sexual partners and sharing shavers, razors or earrings, respectively. In multivariable analysis, history of STDs, having multiple sexual partners, and sharing shavers, razors or earrings were considered as potential risk factors for HIV co-infection. HBsAg positive delivering mothers who had a history of STDs were 9.3 times more likely to have HIV co-infection compared to their counterparts (AOR = 9.3; 95% CI: 1.84–47.1). Mothers who had multiple sexual partners were 5.96 times more likely to have HIV co-infection (AOR = 5.96; 95% CI: 1.074–33.104). The odds of having HBV/HIV co-infection were 5.5 times higher among mothers who had a history of sharing shavers, razors and earrings (at homes, beauty salon or barbershops) (AOR = 5.5;95%CI: 1.014–29.69) (Table 3).

## Prevalence of HCV co-infection among HBsAg positive delivering mothers

Only 3 (1.1%) of the HBsAg positive delivering mothers were co-infected with HCV. All of the HCV co-infected mothers were married. Two (66.7%) were between 25–29 years old, had a primary educational level, were orthodox Christians, and multigravida. However, none of the socio-demographic characteristics were significantly associated with co-infection of HCV (Table 4).

**Table 2. Socio-demographic characteristics and HIV co-infection among HBsAg positive delivering mothers in governmental hospitals in Addis Ababa, 2019–2020.**

| Variable | Category | HIV status | | COR (95%CI) | P value |
|---|---|---|---|---|---|
| | | Positive (9) | Negative (256) | | |
| | | N (%) | N (%) | | |
| Age | <20 | 0 | 4 (1.6) | 1.39 (9.658–2.951) | 0.386 |
| | 20–24 | 2 (22.2) | 59 (23) | | |
| | 25–29 | 6 (66.7) | 109 (42.6) | | |
| | 30–34 | 1 (11.1) | 59 (23) | | |
| | >35 | 0 | 25 (9.8) | | |
| Marital status | Married | 7 (77.8) | 238 (93) | 4.86 (.524–44.9) | 0.164 |
| | Single | 1 (11.1) | 11 (4.3) | 1.57 (.084–29.4) | 0.762 |
| | Divorced | 1 (11.1) | 7 (2.7) | 1 | |
| Educational status | Illiterate | 2 (22.2) | 26 (10.2) | .206 (0.18–2.38) | 0.206 |
| | Primary level | 4 (44.4) | 94 (36.7) | .373 (.041–3.42) | 0.383 |
| | Secondary level | 2 (22.2) | 73 (28.5) | .579 (.051–6.54) | 0.659 |
| | College diploma and above | 1 (11.1) | 63 (24.6) | 1 | |
| Occupation | Government employee | 1 (11.1) | 46 (18) | 5.75 (.488–67.8) | 0.165 |
| | Private employee | 2 (22.2) | 50 (19.5) | 3.13 (.407–24.01) | 0.273 |
| | Self- employee | 1 (11.1) | 27 (10.5) | 3.34 (.283–40.25) | 0.336 |
| | House wife | 3 (33.3) | 117 (45.7) | 4.88 (.756–31.44) | 0.096 |
| | Others | 2 (22.2) | 16 (6.3) | 1 | |
| Religion | Orthodox Christian | 6 (66.7) | 184 (71.9) | 7.68 (.741–79.34) | 0.088 |
| | Muslim | 1 (11.1) | 43 (16.8) | 10.75 (.560–206.4) | 0.115 |
| | Protestant | 1 (11.1) | 25 (9.8) | 6.25 (.322–121.33) | 0.226 |
| | Others | 1 (11.1) | 4 (1.6) | 1 | |
| Gravidity | Primigravida | 2 (22.2) | 104 (40.6) | 2.39 (.488–11.757) | 0.282 |
| | Multigravida | 7 (77.8) | 152 (59.4) | 1 | |

## Risk factors associated with HCV co-infection among HBsAg positive delivering mothers

All (100%) of the HCV co-infected delivering mothers had a history of ear piercings, and 2 (66.7%) had a history of tattooing, female circumcision, sharing shavers, razors, and earrings (at homes, beauty salon or, barber shops), and history of jaundice. About 1 (33.3%) had a history of blood transfusion, previous abortion, surgical procedure, dental procedure, multiple sexual partners, and contact with jaundice patients. Even though the history of blood transfusion, tattooing, home delivery by traditional birth attendants, jaundice, and contact with jaundice patients were candidate variables for multivariate analysis (P<0.25), none of them were significantly associated with HCV co-infection in multivariate analysis (Table 5).

## HIV and HCV co-infection among the HBsAg positive delivering mothers

None of the HBsAg positive delivering mothers were co-infected with both HIV and HCV. None of them were with triplex infection (Table 6).

## Discussion

In the present study, 3.4% and 1.1% of the HBsAg positive delivering mothers were co-infected with HIV and HCV, respectively. There is a scarcity of data on co-infection of HIV or HCV among HBsAg positive delivering mothers, and factors associated to compare with our

**Table 3. Factors associated with HIV co-infection among HBsAg positive delivering mothers in governmental hospitals in Addis Ababa, 2019–2020.**

| Variables | Category | HIV status | | Bivariate analysis | | Multivariate analysis | |
|---|---|---|---|---|---|---|---|
| | | Positive (9) | Negative (256) | P value | COR (95%CI) | AOR (95%CI) | P value |
| | | N (%) | N (%) | | | | |
| History of blood transfusion | Yes | 1 (11.1) | 16 (6.3) | 0.565 | 1.88 (.221–15.93) | NA | |
| | No | 8 (88.9) | 239 (93.7) | | 1 | | |
| History of STDs | Yes | 4 (44.4) | 11(4.3) | 0.000 | 17.82 (4.19–75.7) | 9.3 (1.84–47.1) | 0.007 |
| | No | 5 (55.6) | 246 (95.7) | | 1 | 1 | |
| History of abortion | Yes | 4 (44.4) | 44 (17.2) | 0.051 | 3.86 (.995–14.9) | 2.69 (.539–13.4) | 0.227 |
| | No | 5 (55.6) | 212 (82.8) | | 1 | 1 | |
| History of surgical procedure | Yes | 1 (11.1) | 27 (10.6) | 0.926 | 1.11 (.133–9.19) | NA | |
| | No | 8 (88.9) | 230 (89.4) | | 1 | | |
| History of dental procedure | Yes | 1 (11.1) | 37 (14.5) | 0.779 | .74 (.090–6.1) | NA | |
| | No | 8 (88.9) | 219 (85.5) | | 1 | | |
| History of tattooing | Yes | 2 (22.2) | 44 (17.2) | 0.696 | 1.37 (.277–6.58) | NA | |
| | No | 7 (77.8) | 212 (82.8) | | 1 | | |
| History of ear piercings | Yes | 8 (88.9) | 249 (97.3) | 0.186 | .23 (.025–2.05) | 0.26 (.019–3.51) | 0.308 |
| | No | 1 (11.1) | 7 (2.7) | | 1 | 1 | |
| History of nose piercing | Yes | 1 (11.1) | 12 (4.7) | 0.397 | 2.54 (.294–21.99) | NA | |
| | No | 8 (88.9) | 244 (95.3) | | 1 | | |
| History of home delivery by traditional birth attendants | Yes | 1 (11.1) | 8 (3.1) | 0.226 | 3.88 (.432–34.79) | 0.83 (.042–16.7) | 0.905 |
| | No | 8 (88.9) | 248 (96.9) | | 1 | | |
| History of having multiple sexual partners | Yes | 7 (77.8) | 67 (26.2) | 0.005 | 5.53 (1.345–22.73) | 5.96 (1.074–33.104) | 0.041 |
| | No | 2 (22.2) | 189 (73.8) | | 1 | | |
| History of female circumcision | Yes | 4 (44.4) | 123 (48) | 0.692 | 1.31 (.349–5.00) | NA | |
| | No | 5 (55.6) | 133 (52) | | 1 | | |
| History of hospital admission | Yes | 2 (22.2) | 38 (14.8) | 0.552 | 1.69 (.338–8.46) | NA | |
| | No | 7 (77.8) | 218 (85.2) | | 1 | | |
| Presence of known hepatitis B infected person in a family | Yes | 2 (22.2) | 26 (10.2) | 0.263 | 2.53 (.499–12.81) | NA | |
| | No | 7 (77.8) | 230 (89.8) | | 1 | | |
| History of sharing shavers, razors, or earrings (at homes, beauty salon or, barber shops) | Yes | 6 (66.7) | 72 (28.1) | 0.022 | 5.21 (1.269–21.4) | 5.5 (1.014–29.69) | 0.048 |
| | No | 3 (33.3) | 184(71.9) | | 1 | 1 | |
| History of sharing tooth brushes with others | Yes | 1 (11.1) | 10 (3.9) | 0.311 | 3.08 (.350–27.00) | NA | |
| | No | 8 (88.9) | 246 (96.1) | | 1 | | |
| History of jaundice | Yes | 1 (11.1) | 12 (4.7) | 0.397 | 2.54 (.294–21.99) | NA | |
| | No | 8 (88.9) | 244 (95.3) | | 1 | | |
| History of contact with jaundice patient | Yes | 1 (11.1) | 16 (6.3) | 0.565 | 1.88 (.221–15.93) | NA | |
| | No | 8 (88.9) | 240 (93.7) | | 1 | | |

CI: confidence interval

COR: Crude odds ratio, AOR: adjusted odds ratio

NA: not applicable refers to factors with p-value ≥0.25 at the bivariate analysis which were not considered in Multivariable analysis.

**Table 4. Socio-demographic characteristics and HCV co-infection among HBsAg positive delivering mothers in governmental hospitals in Addis Ababa, 2019–2020.**

| Variable | Category | HCV status | | P value |
|---|---|---|---|---|
| | | Positive (3) | Negative (262) | |
| | | N (%) | N (%) | |
| Age | <20 | 0 | 4 (1.5) | 0.348 |
| | 20–24 | 0 | 61 (23.8) | |
| | 25–29 | 2 (66.7) | 113 (44.1) | |
| | 30–34 | 0 | 60 (23.4) | |
| | >35 | 1 (33.3) | 24 (9.2) | |
| Marital status | Married | 3 (100) | 245 (93.5) | 0.999 |
| | Single | 0 | 12 (4.5) | |
| | Divorced | 0 | 8 (3.1) | |
| Educational status | Illiterate | 0 | 28 (10.9) | 0.997 |
| | Primary level | 2 (66.7) | 96 (37.5) | |
| | Secondary level | 0 | 75 (29.3) | |
| | College diploma and above | 1 (33.3) | 63 (20) | |
| Occupation | Government employee | 0 | 47 (18.4) | 0.888 |
| | Private employee | 1 (33.3) | 51 (19.5) | |
| | Self- employee | 1 (33.3) | 26 (9.9) | |
| | House wife | 1 (33.3) | 119 (46.5) | |
| | Others | 0 | 19 (7.3) | |
| Religion | Orthodox Christian | 2 (66.7) | 188 (73.4) | 0.128 |
| | Muslim | 0 | 44 (17.2) | |
| | Protestant | 0 | 26 (9.9) | |
| | Others | 1 (33.3) | 4 (1.5) | |
| Gravidity | Primigravida | 1 (33.3) | 105 (40.1) | 0.813 |
| | Multigravida | 2 (66.7) | 157 (59.9) | |

findings. This is due to most of the previous studies were conducted on general or HIV-infected pregnant mothers.

The 3.4% HIV co-infection prevalence in our study is similar to studies in South Africa (3.1%) [21] and Rwanda (4.1%) [10], but it was smaller than HBV and HIV co-infection found among pregnant women in Europe (4.9%) [17], Nigeria (11.8%) [20], Bahir Dar city, Ethiopia (19%) [25], Atat Hospital, Southern Ethiopia (40%) [7] and Addis Ababa, Ethiopia (22.2%) [26]. In contrast it was higher than studies reported in Nigeria (0.24–2%) [23, 30], Southern Ethiopia (0.6%) [31] and 0% in eastern Ethiopia [32]. Furthermore, the 1.1% HCV co-infection in this study is similar to the study in Pakistan (1.3%) [33]. But it is lower than a study in west Iran (7.9%) [11] and higher than a study in Ghana (0.6%) [34]. None of the pregnant mothers were co-infected with HBV and HCV in studies in Ethiopia [28], Nigeria [23], Pakistan [27], and Sana'a, Yemen [35]. These variations might be due to differences in sampling method, laboratory tests used, and cultural and behavioural practices. Even though all of the socio-demographic characteristics were not significantly associated with both HIV and HCV co-infections, the majority (66.7%) of the co-infected mothers were in the age group of 25–29 years old. This is in line with the study conducted in Nigeria [30]. Seven (77.8%) of the HIV co-infected and 66.7% of the HCV co-infected mothers were multigravida. This is supported by studies from Rwanda [13] and Nigeria [6].

In our study, the majority of the potential risk factors were not significantly associated with HCV and HIV co-infections. This might be due to the low number of HBV/HIV and HBV/

**Table 5. Statistical association of predictor variables with HCV co-infection among HBsAg positive delivering mothers in governmental hospitals in Addis Ababa, 2019–2020.**

| Variables | Category | HCV status | | Bivariate analysis | | Multivariate analysis | |
|---|---|---|---|---|---|---|---|
| | | Positive (3) | Negative (262) | P value | COR (95%CI) | AOR (95%CI) | P value |
| | | N (%) | N (%) | | | | |
| History of blood transfusion | Yes | 1 (33.3) | 16 (6.1) | 0.103 | 7.68 (.661–89.36) | 1.98 (.045–87.66) | 0.722 |
| | No | 2 (66.7) | 246 (93.9) | | 1 | | |
| History of STDs | Yes | 0 | 15 (5.7) | | | | |
| | No | 3 (100) | 247 (94.3) | | | | |
| History of abortion | Yes | 1 (33.3) | 48 (18.3) | 0.526 | 2.23 (.198–15.08) | NA | |
| | No | 2 (66.7) | 214 (81.7) | | 1 | | |
| History of surgical procedure | Yes | 1 (33.3) | 27 (10.3) | 0.236 | 4.35 (.382–49.6) | | |
| | No | 2 (66.7) | 235 (89.7) | | | | |
| History of dental procedure | Yes | 1 (33.3) | 37 (14.1) | 0.369 | 3.04 (.269–34.38) | NA | |
| | No | 2 (66.7) | 225 (85.9) | | 1 | | |
| History of tattooing | Yes | 2 (66.7) | 42 (16) | 0.057 | 10.48 (.929–118.2) | 4.23 (.222–80.79) | 0.338 |
| | No | 1 (33.3) | 220 (84) | | 1 | | |
| History of ear piercings | Yes | 3 (100) | 254 (96.9) | | | | |
| | No | 0 | 8 (3.1) | | | | |
| History of nose piercing | Yes | 0 | 13 (5) | | | | |
| | No | 3 (100) | 249 (95) | | | | |
| History of home delivery by traditional birth attendants | Yes | 1 (33.3) | 8 (3.0) | 0.030 | 15.88(1.30–193.68) | 10.92 (.179–665.05) | 0.254 |
| | No | 2 (66.7) | 254 (97) | | 1 | | |
| History of having multiple sexual partners | Yes | 1 (33.3) | 73 (27.9) | 0.834 | 1.3 (.116–14.5) | NA | |
| | No | 2 (66.7) | 189 (72.1) | | 1 | | |
| History of female circumcision | Yes | 2 (66.7) | 99 (37.8) | 0.333 | 3.293 (.295–36.79) | NA | |
| | No | 1 (33.3) | 163 (62.2) | | 1 | | |
| History of hospital admission | Yes | 1 (33.3) | 38 (14.5) | 0.382 | 2.95 (.261–33.3) | NA | |
| | No | 2 (66.7) | 223 (85.5) | | 1 | | |
| Presence of known hepatitis B infected person in a family | Yes | 0 | 28 (10.7) | | | | |
| | No | 3 (100) | 234 (89.3) | | | | |
| History of sharing shavers, razors or earrings (at homes, beauty salon or, barber shops) | Yes | 2 (66.7) | 75 (28.6) | 0.192 | 4.99(.445–55.32) | 6.23(.292–133.3) | 0.242 |
| | No | 1 (33.3) | 187 (71.4) | | 1 | | |
| History of sharing tooth brushes with others | Yes | 0 | 11 (4.2) | | | | |
| | No | 3 (100) | 251 (95.8) | | | | |
| History of jaundice | Yes | 2 (66.7) | 11 (4.2) | 0.002 | 45.64 (3.84–542.4) | 25.16 (.979–646.82) | 0.052 |
| | No | 1 (33.3) | 251 (95.8) | | 1 | | |
| History of contact with jaundice patient | Yes | 1 (33.3) | 16 (6.1) | 0.103 | 7.69 (.661–89.37) | 3.22 (.104–99.43) | 0.504 |
| | No | 2 (66.7) | 246 (93.9) | | | | |

COR: crude odds ratio, AOR: Adjusted odds ratio

CI: confidence interval

NA: not applicable refers to factors with p-value ≥0.25 at the bivariate analysis which were not considered in Multivariable analysis.

**Table 6. HIV and HCV co-infection among HBsAg positive delivering mothers in governmental hospitals in Addis Ababa, 2019–2020.**

| HIV status | HCV status | | p-value |
|---|---|---|---|
| | **Positive** | **Negative** | |
| Positive | 0 | 9 | 0.999 |
| Negative | 3 | 253 | |

HCV co-infected study participants in our study, HBV being infectious than HIV and HCV, or early childhood transmission of HBV rather than shared routes of HIV, HCV, and HBV in adulthood [10, 21].

History of having STDs was significantly associated with HIV co-infection. This is in agreement with studies in Rwanda [10] and Brazil [36]. This might be due to STDs being able to increase the susceptibility to infection by both viruses through mucosal disruption, immune changes, and microenvironment effects on the genital tract. History of having multiple sexual partners was another significant risk factor for the HBV/HIV co-infection. This is in agreement with studies in Southern Ethiopia [7], Rwanda [10], and Brazil [36]. This might be due to both HIV and HBV are sexually transmitted infections, and the transmission risk increases with the number of sexual partners. The history of sharing shavers, razors, and earrings (at homes, beauty salons, or barbershops) was also another significant risk factor to HIV co-infection in our study. This is similar to a study in Indonesia, which identified that HIV transmission is possible due to lack of cleanliness of shaving equipment [37]. This might be due to HIV and other blood born infections can be transmitted via the use of sharing sharp material.

## Conclusion and recommendation

Of the HBsAg positive delivering mothers, 3.4% and 1.1% were dual infected with HIV and HCV, respectively. This indicates that a substantial number of infants born in Ethiopia are at high risk of mother to child transmission of HBV, HIV and HCV. All of the socio-demographic characteristics of the study participants and the majority of the potential risk factors were not significantly associated with co-infection of the viruses. History of having STDs, having multiple sexual partners, and sharing of shavers, razors, and earrings were potential risk factors significantly associated with co-infection of HIV. All pregnant mothers need to be screened for HBV, HCV, and HIV during antenatal care and also need implementation of prevention mechanisms of MTCT of those viral infections. Furthermore, awareness creation on the transmission of those blood born viral infections is needed.

## Strengths and limitations of the study

We included a large number of HBsAg positive delivering mothers than previous studies and, it can help to more determine the rate and potential risk factors of the co-infections. Confirmatory tests like ELISA and PCR were not used because of a lack of laboratory setup.

## Supporting information

**S1 File.**
(DOCX)

## Acknowledgments

We gratefully acknowledge the study participants, data collectors, and supervisors who participated in data collection.

## Author Contributions

**Conceptualization:** Mebrihit Arefaine Tesfu, Nega Berhe Belay, Tilahun Teklehaymanot Habtemariam.

**Data curation:** Mebrihit Arefaine Tesfu.

**Formal analysis:** Mebrihit Arefaine Tesfu.

**Investigation:** Mebrihit Arefaine Tesfu.

**Methodology:** Mebrihit Arefaine Tesfu, Nega Berhe Belay, Tilahun Teklehaymanot Habtemariam.

**Supervision:** Mebrihit Arefaine Tesfu.

**Validation:** Mebrihit Arefaine Tesfu, Nega Berhe Belay, Tilahun Teklehaymanot Habtemariam.

**Writing – original draft:** Mebrihit Arefaine Tesfu.

**Writing – review & editing:** Mebrihit Arefaine Tesfu, Nega Berhe Belay.

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
