## [Decision Letter · Decision Letter 0]

20 Jul 2021

PONE-D-21-13799

Co-infection of HIV or HCV among HBsAg positive delivering mothers and its associated factors in governmental hospitals in Addis Ababa, Ethiopia: a cross-sectional study

PLOS ONE

Dear Mebrihit Arefaine Tesfu,

Thank you for submitting your manuscript to PLOS ONE. After careful consideration, we feel that it has merit but does not fully meet PLOS ONE’s publication criteria as it currently stands. Therefore, we invite you to submit a revised version of the manuscript that addresses the points raised during the review process.

Reviewer #1: This study dertermines co-infection of HIV, or HCV and related risk factors among HBsAg positive delivering mothers in Ethiopia .HBV, HCV or HIV infection are major public health problems globally, with more than 1.2 million HBV infection-related deaths annually. Many previous studies have explored the co-infection of HBV-HIV, HBV-HCV and HIV-HCV among the general population, general pregnant mothers and pregnant mothers infected with HIV. This study chose another perspective to explore the co-infection of HIV or HCV in Ethiopian among HBsAg positive delivering pregnant mothers. and analyze its risk factors.

This research method is reasonable, the results are clear and complete, the conclusion shows the infection rate of the co-infection of HIV or HCV among HBsAg positive delivering mothers, and risk factors. The suggestions are provided for the clinical, if these viral infection rate of the babes from these delivering pregnant women can be given, that would be more complete and prefect.

Reviewer #2: Overall, I think the paper provides information needed in the field. It just needs some major editing and formatting to make it more presentable and easier to read for the general public.

General Comments

1. There are several parts in the tables where they need to be properly formatted. There are sections of the tables where spacing is off between numbers and percentages.

2. Fix abbreviations throughout the paper. When referring to Hepatitis B Virus, Hepatitis C Virus, and Human Immunodeficiency Virus, you should spell them out at first use in the paper and put the abbreviation in parentheses right after the first spelled out. Then use the abbreviation throughout the rest of the paper. This could also be done to abbreviate mother-to-child-transmission as MTCT.

3. Need to consistently use either "HBV-positive" or "HBV positive" throughout the paper. They are both used but one style of writing that needs to be used. This applies to HIV-positive, HCV-positive, and HBsAg-positive as well.

4. Be sure to indent the beginning of each paragraph throughout paper.

5. Check the formatting of reference list and in-text citations to ensure they adhere to journal guidelines.

6. Be sure to put in-text citations at the end of the sentence, not randomly in the middle.

Specific Comments:

I have attached the PDF of the manuscript with specific comments throughout so that all my comments can be easily seen.

Words that are marked through with a red line are ones I think should be deleted.

ACADEMIC EDITOR:

Typographic errors should be addressedProvide specific feedback from your evaluation of the manuscript

Please submit your revised manuscript by 6th August 2021.  If you will need more time than this to complete your revisions, please reply to this message or contact the journal office at plosone@plos.org. Please include the following items when submitting your revised manuscript:

We look forward to receiving your revised manuscript.

Kind regards,

Edford Sinkala

Academic Editor

PLOS ONE

Additional Editor Comments:

Please attend to the many typographic errors

Journal Requirements:

2. Please include additional information regarding the survey or questionnaire used in the study and ensure that you have provided sufficient details that others could replicate the analyses. For instance, if you developed the survey or questionnaire as part of this study and it is not under a copyright more restrictive than CC-BY, please include a copy, in both the original language and English, as Supporting Information. If the questionnaire is published, please provide a citation to the (1) questionnaire and/or (2) original publication associated with the questionnaire.

Reviewers' comments:

Reviewer's Responses to Questions

**Comments to the Author**

1. Is the manuscript technically sound, and do the data support the conclusions?

Reviewer #1: Yes

Reviewer #2: Yes

2. Has the statistical analysis been performed appropriately and rigorously? 

Reviewer #1: Yes

Reviewer #2: Yes

3. Have the authors made all data underlying the findings in their manuscript fully available?

Reviewer #1: Yes

Reviewer #2: Yes

4. Is the manuscript presented in an intelligible fashion and written in standard English?

Reviewer #1: Yes

Reviewer #2: No

5. Review Comments to the Author

6. PLOS authors have the option to publish the peer review history of their article (what does this mean?). If published, this will include your full peer review and any attached files.

Reviewer #1: **Yes: **Binyang Luo

Reviewer #2: **Yes: **Sarah Gayle Franklin

---

## [Author Response · Author response to Decision Letter 0]

6 Aug 2021

We appreciate the time and effort that the Editor and reviewers dedicated to providing feedback on our manuscript and are grateful for the insightful comments on and valuable improvements to our paper. We have incorporated almost all of the comments made by the Editor and reviewers. Those changes are highlighted within the revised manuscript.

---

## [Decision Letter · Decision Letter 1]

9 Jun 2022

PONE-D-21-13799R1Co-infection of HIV or HCV among HBsAg positive delivering mothers and its associated factors in governmental hospitals in Addis Ababa, Ethiopia: a cross-sectional studyPLOS ONE

Dear Dr. Tesfu,

Thank you for submitting your manuscript to PLOS ONE. After careful consideration, we feel that it has merit but does not fully meet PLOS ONE’s publication criteria as it currently stands. Therefore, we invite you to submit a revised version of the manuscript that addresses the points raised during the review process.

The reviewers overall felt that your revised manuscript has improved. However, they raised a few minor points which must be addressed. These include the reformatting of some of your references, which may contain errors. Please see the reviewers' comments below.

We look forward to receiving your revised manuscript.

Kind regards,

Natasha McDonald, PhD

Associate Editor

PLOS ONE

Journal Requirements:

Reviewers' comments:

Reviewer's Responses to Questions

**Comments to the Author**

1. If the authors have adequately addressed your comments raised in a previous round of review and you feel that this manuscript is now acceptable for publication, you may indicate that here to bypass the “Comments to the Author” section, enter your conflict of interest statement in the “Confidential to Editor” section, and submit your "Accept" recommendation.

Reviewer #2: All comments have been addressed

Reviewer #3: All comments have been addressed

Reviewer #4: All comments have been addressed

Reviewer #5: All comments have been addressed

2. Is the manuscript technically sound, and do the data support the conclusions?

Reviewer #2: Yes

Reviewer #3: Yes

Reviewer #4: Yes

Reviewer #5: Yes

3. Has the statistical analysis been performed appropriately and rigorously? 

Reviewer #2: N/A

Reviewer #3: Yes

Reviewer #4: Yes

Reviewer #5: Yes

4. Have the authors made all data underlying the findings in their manuscript fully available?

Reviewer #2: Yes

Reviewer #3: No

Reviewer #4: Yes

Reviewer #5: Yes

5. Is the manuscript presented in an intelligible fashion and written in standard English?

Reviewer #2: Yes

Reviewer #3: No

Reviewer #4: Yes

Reviewer #5: Yes

6. Review Comments to the Author

Reviewer #2: Authors have addressed previous comments by reviewers and have improved their paper significantly. I would agree that it should be forwarded to publication.

Reviewer #3: This is an interesting study.

The authors should provide key words at the end of the abstract.

The authors should include the prevalence of cases with triplex infections.

REFERENCES

Some references were wrongly done. Example, reference 24 should be: Duru MU, Aluyi HS, Anukam KC. Rapid screening for co-infection of HIV and HCV in pregnant women in Benin City, Edo State, Nigeria. Afr Health Sci. 2009 Sep;9(3):137-42. PMID: 20589140; PMCID: PMC2887022.

Reviewer #4: (No Response)

Reviewer #5: Interesting results for Ethiopia which is important in designing prevention programmes from mother-to-child of HIV, HBV and HCV. The team may extend the studies to other parts of Ethiopia because birth practices may differ.

7. PLOS authors have the option to publish the peer review history of their article (what does this mean?). If published, this will include your full peer review and any attached files.

Reviewer #2: No

Reviewer #3: **Yes: **George Eleje

Reviewer #4: **Yes: **Moses P. Adoga

Reviewer #5: No

---

## [Author Response · Author response to Decision Letter 1]

24 Jul 2022

 Author response: Thank you for pointing this out. We accepted the comment, and we have reviewed the reference list. We found that references number 27 and 29 were retracted papers and replaced with other current references. References number 3 and 24 were wrongly done and corrected in the revised manuscript. 

2. The authors should provide key words at the end of the abstract (Reviewer 3) 

 Author response: Based on the submission guideline format of PLOS ONE journal, keywords should not be included at the end of the abstract. But, they incorporate during online submission. That is why we cannot provide keywords at the end of the abstract.

3. The authors should include the prevalence of cases with triplex infections.

 Author response: we accepted the reviewers’ comment and included in the manuscript.

4. REFERENCES

Some references were wrongly done. Example, reference 24 should be: Duru MU, Aluyi HS, Anukam KC. Rapid screening for co-infection of HIV and HCV in pregnant women in Benin City, Edo State, Nigeria. Afr Health Sci. 2009 Sep;9(3):137-42. PMID: 20589140; PMCID: PMC2887022.

Author response: we accept the comment and corrected it.

---

## [Editor Report · Decision Letter 2]

8 Aug 2022

Co-infection of HIV or HCV among HBsAg positive delivering mothers and its associated factors in governmental hospitals in Addis Ababa, Ethiopia: a cross-sectional study

PONE-D-21-13799R2

Dear Dr. Tesfu,

We’re pleased to inform you that your manuscript has been judged scientifically suitable for publication and will be formally accepted for publication once it meets all outstanding technical requirements.

Kind regards,

Hugh Cowley

Staff Editor

PLOS ONE

Additional Editor Comments (optional):

When finalizing your manuscript for publication, please check the newly added reference 29 and ensure that enough information is given to enable a reader to locate with certainty the literature you are referencing. Please see our submission guidelines for guidance: https://journals.plos.org/plosone/s/submission-guidelines#loc-references.
---

## [Editor Report · Acceptance letter]

17 Aug 2022

PONE-D-21-13799R2 

Co-infection of HIV or HCV among HBsAg positive delivering mothers and its associated factors in governmental hospitals in Addis Ababa, Ethiopia: a cross-sectional study 

Dear Dr. Tesfu:

I'm pleased to inform you that your manuscript has been deemed suitable for publication in PLOS ONE. Congratulations! Your manuscript is now with our production department. 

Kind regards, 

on behalf of

Ms. Natasha McDonald 

%CORR_ED_EDITOR_ROLE%

PLOS ONE